# Business Models and Innovation in the Indonesian Smallholder Beef Value Chain

**Zenal Asikin** [1,2,*] , **Derek Baker** [1] , **Renato Villano** [1] **and Arief Daryanto** [2]

1   UNE Business School, University of New England, Armidale, NSW 2351, Australia;
    Derek.Baker@une.edu.au (D.B.); rvillan2@une.edu.au (R.V.)
2   School of Business, IPB University (Bogor Agricultural University), Bogor 16151, Indonesia;
    adaryant@yahoo.com
*   Correspondence: zasikin@apps.ipb.ac.id

**Abstract:** This paper proposes a framework for identification of business models in smallholder cattle production and marketing that represents innovation behaviour. Cattle are vital to Indonesia's smallholders' livelihoods, and smallholder systems are key to serving the country's growing demand for beef. Business incentives currently select against the development and sustainability of breeding systems which would support domestic production, and so new models are needed which utilize innovation. The three primary components of a business model are presented: its value proposition, its value architecture, and its financing mechanism. A research approach is provided, by way of mapping data needs to the business models, and proposing relationships between observed innovation practices and the business models within the value chain. For implementation, the paper provides guidance on facilitation needs and the role of stakeholders in the case of the Indonesian cattle and beef value chain.

**Keywords:** business models; innovation behaviour; value proposition; value architecture; financing mechanism; novel framework proposal; smallholder farming systems; Indonesian cattle and beef value chain; role of stakeholders

## 1. Introduction

Indonesian demand for beef has exhibited recent strong growth [1], due to increasing middle-class income and population growth [2–5]. Java Island, with 50% of the country's population, contributes more than 70% of total beef consumption [6]. Indonesian consumption patterns are also shifting away from staple foods toward high-value agriculture products (HVAPs) [3,4]. Most (90%) Indonesian beef producers are smallholders [5,7]. They provide some 65–70% of all domestic beef production [8,9]. Smallholders' cows produce almost all (98%) of the country's calves, mostly in integrated crop and livestock farming systems [10]. Hence, government goals regarding establishing a balance between domestic supply and demand must address smallholders.

Beef production has not kept pace with demand growth [1,3]. Government actions to increase beef production include promoting the country's cow and calf production system. As Indonesia lacks large open-grazing lands and is densely populated, intensive land use means that transported feed is a significant input to cattle production and its cost is a significant determining factor of both production and income from beef [11]. Indonesia's numerous crops that could provide cheap sources of cattle feed for both breeding and fattening mean that crop-livestock integration is a potential feeding solution. More feed and better management would also address productivity.

Smallholders face constraints to improvement of their production and marketing systems. One such constraint relates to the lack of quality and quantity of forage, especially in the dry session [11–14].

Most of the feed sources such as grazing, cut grass, and crop residues are from on-farm resources. There is a modest and localised feed market, but traded feed is considered high cost [15]. Feed costs constitute some 60% of all production costs, generally split between cash cost (10%) and non-cash cost (50%) [16].

In terms of innovation to overcome these constraints, Indonesian smallholders' adoption of technology to improve beef productivity has been slow [12]. Similarly, poor practices, including feed management, are widely observed [4,5,12]. Market access is another constraint [12] along with a lack of coordination and weak value chain linkages [11]. This reduces options for marketing. A further constraint relates to lack of access to finance [4,12,17], which results in some feedback effects such as smallholders' not being able to buy bulls and cows [11,12], or machinery.

The contribution of this paper is the provision of a framework to identify business models in smallholder cattle production and marketing that represent innovation behaviour. The paper begins with an overview of the Indonesian cattle and beef value chain, followed by a discussion of the role of business models in a sustainable commercial beef industry, particularly in addressing the constraints faced. A business model definition and characterisation are identified from the literature. Mapping of data needs to business models, and relationships between innovation practices and business models, are developed. Finally, yet importantly, facilitation needs, and roles for stakeholders in developing business models, are presented in the specific context of Indonesian smallholder beef producers. Conclusions and research directions are also offered.

## 2. The Indonesian Cattle and Beef Value Chain

Livestock production in Indonesia is one part of a broader set of small-scale integrated farming systems. Animal feed is comprised not only of forage and crop residues but also additional feeds such as rice bran, cassava starch, wheat pollard, or coconut meal [9], generally fed without concentrates [18]. Most husbandry activities are rudimentary [11], and small land parcels are employed for the diverse enterprises of crop, fodder, livestock, and household activities [12,19].

Crops generally fulfil household needs, with surpluses sold or bartered, and livestock income is secondary or used in an emergency [19]. Hence, smallholder livestock farmers have been depicted as "users" and "keepers" rather than "producers" [4,19]. Smallholders have small numbers of animals (2–5 head) which, in the absence of crop-livestock integration, generally graze in backyards, on roadsides, and in local forests [19]. Productivity is consequently low [12,17]. However, across income, savings and cultural values, beef plays a vital role in smallholder livelihoods [3,11,17]. Indonesian beef farming system types include breeding, fattening and combinations of both breeding and fattening. Most livestock-keeping smallholders (76%) operate breeding, followed by those fattening (24%) [7]. While there are no official data for the combination of breeding and fattening, a number of households do both. Selling decisions are based on household needs, and so livestock serves as a savings function [8], and in general, the main purpose of breeding systems is not regular income generation [18]. In the fattening systems, however, regular income is the main objective [18].

Livestock production contributed 12% of total 2014 agricultural household income and was the third most important income-generating activity after estate crops (33%) and food crops (32%) [7]. For those households with livestock as the main source of income, livestock production accounted for 75% of household income, followed by rice (8%), secondary crops (4%), estate crops (5%), horticulture (4%), and forestry (3%). Livestock also contributed 4% of the income of estate crop households, 6% of horticultural household income, and 13% to households growing food crops as a main activity [20].

Scholars report that, in Indonesian smallholder beef farming, fattening systems have the highest profitability followed by combinations of breeding and fattening, and that breeding has the lowest [8,10]. Private investors have not been interested in cattle breeding because of the low, and slow, return available [21] and the long investment period [22,23]. In the medium- and large-scale farms, fattening for *qurban* (on the occasion of 'Eid Al-Adha) shows higher profitability than does sale to slaughterhouses

due to the prices received [8]. However, a sustainable fattening system requires young cattle produced by a viable breeding system.

Smallholders also have been observed switching from breeding to fattening due to constraints on grazing land [8]. Smallholders obtain forage mainly by cut-and-carry activities, which are time-intensive and primarily utilise family labour. If family labour costs are included, profitability is seen to be further reduced [8]. Crop-livestock integration has been shown to be profitable [8], and semi-intensive systems may be more profitable than intensive ones. Government release of land for cattle grazing, and favourable credit conditions for feed purchase or access, have been proposed [10].

Currently, many actors in the beef marketing chain in Indonesia supply local butchers and the interisland trade [11,15]. Most smallholders sell their cattle through spot markets which employ ad hoc judgements of animals' attributes, rather than weight. Along with the presence of intermediary actors such as brokers and collectors [3], this influences marketing efficiency [15]. Particularly in the (dominant) spot-cattle marketing system, intermediaries put downward pressure on the cattle prices received by farmers. However, overall, the market power is concentrated at the retail end of the cattle supply chain, particularly as it applies to breeding cattle [11]. Intermediary actors generally dictate cattle prices along the value chain, and buyers and sellers do not negotiate directly [3]. There is no history of collaboration amongst smallholders in cattle marketing. According to [15], the spot marketing system is, generally, a low-cost transaction system with high volume and good access, and good information generation for farmers, but it does not operate efficiently.

Indonesia's end consumers of beef buy meat in three retail formats: wet markets; modern retail; and butchers' shops [11]. Wet markets, with the majority of customers being households, sell most of the beef (60%), followed by stalls selling ready-to-eat meatball and soup dishes (30%), then restaurants and supermarkets (10%) purchasing at wholesale [17]. The modern retail format sells a small amount of beef: currently, it has a market share of around 7%, although this is projected to expand [6].

Smallholders' beef tends to end up in the wet market, while imported beef is utilised in the food service sector and is sold in modern retail [2] mainly at supermarkets and hypermarkets (62% of imported beef) [6]. Some sell in the wet market (20% of imported beef), butcheries (17%), and online retailers (2%) [6]. Indonesia imports of live animal and frozen meat are approximately 30% of national consumption [18,24], mainly from Australia and New Zealand [2,24]. These countries had a market share of 81% and 15% in 2016, respectively [2]. Most imports from Australia are in the form of live cattle for slaughter, but others are for breeding, and for fattening [25]. Fattening of Australian feeder cattle occurs in commercial feedlots [1,17], and more than 70% of Australian-origin beef is consumed in the Greater Jakarta region [6].

Increased competition from Indian buffalo meat, and increasing complexity of regulation, led to a significant decline in imports of Australian cattle and New Zealand meat to Indonesia, by about 7% and 9% on average, respectively, in 2018 [6]. In 2018, Indonesia's source of imported meat, both fresh and frozen, were about 49% from India, 42% (Australia), 3% each from the United States, and New Zealand, as well as from other countries such as Spain, Canada, and Singapore [26].

Commercial feedlots provide beef to Greater Jakarta's markets [1,6]. They operate, generally, in West Java and Lampung, Southern Sumatera [1]. Imported Australian cattle pass through the main ports: Tanjung Priok port in Jakarta and Panjang port in Lampung [1]. Some commercial feedlots have advanced slaughterhouses, so they process their own beef [1,3] and lease kill space to local butchers [3]. Most feedlots work with private distributors and traditional traders in the wet market, to distribute the product [1]. In the 1990s, commercial feedlots and smallholder farmers had partnerships by way of the nucleus estates and smallholders' (NES) approach [17] using imported cattle. However, since the Global Financial Crisis, this business model has almost ceased operation due to the rupiah's depreciation [17]. Today, a few commercial feedlots in Lampung and East Java have contract farming and supply arrangements with smallholders.

Indonesia features a seasonal pattern of demand for cattle and beef, following religious festivals like Ramadan, 'Eid Al-Fitr, 'Eid Al-Adha, Christmas [1,3,11,16,27], and school holidays [16,27]. This pattern

influences the price of beef [1,3], which in turn influences slaughter rates including for female stock (including pregnant cows), and young bulls [1,28]. This configuration of slaughter influences the weights reported and overall productivity. A further observation on seasonal issues is that access to breeding bulls has been limited [29], which affects the sustainability of supply and has resulted in calls for adjustment of the seasonal breeding cycle [15].

## 3. Business Models

The term business model apparently first appeared in an academic article in 1957 [30]. Subsequent literature related to business models is very fragmented because of authors' disparate perspectives and definition. For the purposes of this paper, the business model is the mechanism by which companies do business and generate revenue [31–33]. Hence, we refer to how chain actors and activities are organised [34,35] so as to generate greater value [32], and thus create and retain value and ensure profitability [36], productivity, and sales [37]. Hence, the main principles of a business model relate to market-oriented activities [38,39] to provide win–win solutions for multiple actors in the value chain [39].

Market orientation is important in applying this definition to smallholders, with regard to inclusivity. A business model is said to be inclusive when it entails smallholders' engagement with markets [40]. This improves the development and performance of supply chain relationships, and coordination amongst supply chain actors [40], targeting a win–win situation for both smallholders and buyers [40,41]. Dahlanuddin et al. [42] have observed that specific value-adding actions by smallholder cattle producers in Sumbawa have resulted in collaborative outcomes along the value chain with higher returns to both intermediaries and smallholder producers.

A new business model performs well if it serves customer needs within its business environment, and thus it can facilitate and represent a form of innovation [43]. Teece [43] stated that choosing, changing and/or developing a business model is an intricate art. Good business model design is then highly situational, and the model development process tends to be iterative: arrived at by trial and error.

Spieth and Schneider [44] identified three functions of business models: to describe a way of doing business; to facilitate opportunity development; and to commercialize new ideas and technologies. The first function refers to descriptions of the revenue mechanism, and the architecture generating revenue and profit. The second function concerns facilitative activities and opportunity creation and recognition. The third is a core function linking business activities to both innovation and value creation.

Received research variously identifies from three to nine elements of a business model. These centre on the source of revenue, the product and service delivery, and the value proposition as a central business idea [32]. Richardson [32] proposed three components of a business model framework: its value proposition; its value creation delivery system; and its value capture. The value proposition is also seen as the "nucleus" of a sustainable business model [45]. These dimensions are also applied by scholars such as Perić, Vitezić and Đurkin [38], Remane et al. [46], Sousa-Zomer and Cauchick Miguel [47], Bocken et al. [48], Bocken and Short [49], Yang et al. [50], to address broader business model components. Other scholars as cited by Spieth and Schneider [44] identified three major components, using slightly varying terminology but similar standpoints: a value offering, a value architecture, and revenue model. A systematic literature review by Barth et al. [51] for business models in the agri-food sectors employs a similar framework. From the research perspective, it is of particular interest to understand the value intention of the producers, and their attitudes to change and innovation [51].

The value offering, and the targeting of customers, is the basic strategy to achieve competitive advantage. Value creation and delivery, or the value architecture, refers to how to deliver value to these customers. Such activities require resources and capabilities, organisational structures, and positioning within the value network [32,44]. A business model's value architecture includes the distribution channel and the relationships established [52]. Finally, value capture or the revenue model is how profit is generated in terms of revenues and costs [32,44]. Demil and Lecocq [53] and Osterwalder,

Pigneur and Tucci [52] refer to a financial model rather than a revenue model. The term provides a broader perspective so as also to encompass financial planning for flows of capital [44] including investment [52]. Zhang [54] refers to this as the financing mechanism for the business model.

Access to finance has been recognised as one of the primary constraints on the modernisation of smallholder agriculture [34]. For smallholder cattle and beef in Indonesia, this refers to financing mechanisms, not only in the short term for financing production items like feed, but also in the longer term for investing in capital items like cows or machinery. In this study, the financing mechanism is used as the third functional component of a business model, rather than inserting its financial and revenue roles as support mechanisms to the business model. New business models for smallholder beef producers then feature facilitation of a value proposition, a value architecture and a financing mechanism.

### 3.1. Value Proposition

The term "value proposition" has broad currency in Economics [38] and represents products or services that are offered and delivered to customers: what the producers will provide to their customers, and customer's motivation of customers in purchasing it, and at what price [32].The value proposition should indicate utilitarian value [55], and meeting customer needs [56] to the extent that customers should receive benefits or reduce sacrifices net of associated costs [55]. The customer value proposition both enables the producer's utilisation of resources and capabilities more efficiently than is the case for competitors and mobilises the identification of uniqueness of product or service [55].

Perceived value is comprised of perceived benefit and costs [57]. Value proposition could then refer to one or more of better quality, lower cost, timely delivery, lower price [32,55,56,58–60], to name a few. Its form could be as product or service, as price, or distribution method [61]. Hence, the value proposition plays a fundamental role in business strategy [58] to identify opportunities to create value, and deliver it by way of associated supply chain relationships [62]. Such relationships, in turn, involve shared value for stakeholders [45]. Formulation of the value proposition entails (1) identifying all benefits offered to the customer and (2) identifying points of difference with competitors [63]. The value proposition might take different forms amongst value chain members and along the supply chain.

### 3.2. Value Architecture

The value architecture is the set of activities that produces, sells and delivers the value to customers. Its development employs resources and capabilities, organisation (of the value chain, activity system, and business process) and positioning within value networks (links to suppliers and buyers, and distribution channels used) [32]. Value architecture also refers to governance of transactions [64], and a bulk of research associates the business model with the value chain concept [33] in terms of strategic linkages between value chain actors [65].

In the livestock system, value chain approaches play a significant role due to the significance and complexity of stakeholders' networks, supply chain relationships and governance, and incentive structures [66]. Value chain upgrading is effectively investment in horizontal and vertical aspects: changes in the coordination mechanisms and governance structures [67], as well in the value proposition [68].

### 3.3. Financing Mechanisms

The business model's financing mechanisms include payment and investment flows, from all sources [54,69]. In smallholder livestock systems, the cropping enterprise acts as one source of capital: upgrades to the financing mechanism may affect the value chain's cash flow arrangements for both crops and cattle, and may be oriented internally or externally [35,52,70].

Internal value chain financing occurs, for example, when a trader provides credit to a smallholder, or a firm advances funds to intermediaries for purchases. External sources refer to financing from beyond the chain's boundaries [35]; for example, a financial institution's release of a loan to a

smallholder based on a contract with a buyer [35]. As an upgrade or innovation possibility, credit available to buy animals from smallholders and sell into feedlots would encourage smallholders' market participation.

The foregoing principles are then applied to Indonesian smallholder beef producers. Table 1 maps the business model functions to these stakeholders in terms of value proposition, value architecture and financing mechanism. Changes to the business model components constitute innovations [43] and are presented as value chain upgrades.

**Table 1.** Business model for Indonesian smallholder beef producers.

| | Component of Business Model | | |
| --- | --- | --- | --- |
| | **Value Proposition** | **Value Architecture** | **Financing Mechanism** |
| Features | • Products or services that are offered and delivered to customers | • Organisation and governance in the chain | • Cash flow and payments |
| Functions | • Customer needs met: quality, timing of delivery, price, or combination of these | • Arrangements for production, sales and delivery | • Payments for inputs<br>• Capital sources for investment<br>• Rewards to risk takers |
| Innovation | • Upgrades to change products or services | • Upgrades to change product, service or input delivery | • Upgrades to change payment systems |

## 4. Mapping of Cattle and Beef Value Chains to Business Models by Way of Innovation Practice

Innovation constitutes application of new knowledge, new technology, or new methods [71]. Innovation is also a process of change in production and marketing that may or may not be driven by research [72]. Innovation can also take a variety of forms such as new technology's application to new products, new methods, new markets, or new organisational forms [73]. Innovation is something perceived as new for individuals or organisations, even though it may not be new for others [74]. Innovation related to product and process is referred to as technological innovation, while innovation associated to marketing and organisation is non-technological [75–77]. New business models have been described in terms of innovation practices [43], and we propose here that business models in smallholder cattle and beef value chains can be identified from observed innovation behaviour in production and marketing.

Innovation practices in the cattle and beef value chain may occur across the spectrum of product, process, marketing and organisation. We observe and identify innovation practices as novelties applied by any smallholder, regardless of its newness to other smallholders. A list of candidate innovations includes innovations implemented and the intention of innovation, and the associated smallholder attitudes to change [51]. A variety of innovation practices in cattle production and marketing may occur in terms of marketing, in feeding, breeding, animal health and in collaborative or group actions. Innovations associated with animal feeds include legumes and fodder trees [78]. Improved animal genetic through adoption of crossbreeding contributes to increased meat production [79]. Innovation related to animal health includes cattle surveillance and activity monitors to detect oestrus [80]. Understanding such innovation, and mapping them to business models, requires data such as indicators of the benefit from an innovation, the stakeholders involved, and the enabling factors in making the innovation work (skills, technology, information, assistance, and attitudes to innovation and change).

The general framework to describe the relationship between innovation practices and business models is presented in Figure 1. Study of the business models also requires indicators of the value proposition, value architecture, and financing mechanism. In the marketing system, for example, we should identify the criteria considered by the buyer when buying cattle. Some buyers may

consider timing of delivery and size, while others may use liveweight, breed, or the age of the cattle. This information offers guidance to identify the customer value proposition. Further, information about market conditions and distribution channels, relationships with buyers (vertical relationships), and with other farmers and/or a farmer group (horizontal relationships) would contribute to identifying the business model's value architecture. Finally, yet importantly, information about smallholder's crop-livestock interaction requires observation: in the conventional sense of joint outputs and interactions amongst physical inputs and outputs, but also in terms of funds flows and risk management, and types of payment and transaction used. Reasons for sale and purchase of cattle by smallholders would further contribute to the identification of financing mechanisms. Based on an entrepreneurial view, innovation practices would aim to enhance profits and/or reduce costs [81], and so improve performance [73]. Application of business models as innovations in Indonesian cattle and beef value chains would then similarly aim to increase a range of performance measures such as productivity, cost and risk, and thus profitability. For simplicity, we employ profitability as a measure of performance of a business model, and consequently require revenue and cost information to be collected.

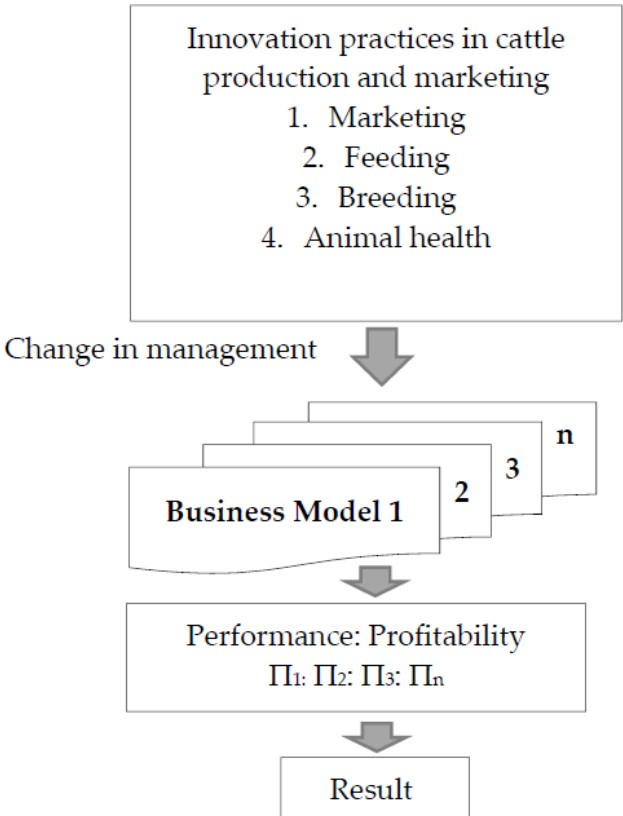

**Figure 1.** Relationship between innovation practices and business model.

## 5. Facilitation Needs for Development of Business Models in Indonesian Cattle and Beef Value Chains

Smallholder beef producers are the intended end users and beneficiaries of innovations, as they apply a new business model with the purpose of, as discussed above, increasing profitability. Intermediary actors in the chain such as brokers, collectors, and informal slaughterers (home slaughterhouse operators) may influence such performance. A challenge for innovation is that intermediaries may be affected significantly, and so its design would ideally include their participation.

Development of business models for innovation by smallholder cattle producers requires facilitation. As a result, support from organisations responsible for livestock extension and research is

desirable. Further, policies conducive to innovation uptake will also contribute to development of business models.

## 6. Roles for External Stakeholders

Beyond smallholder producers and value chain participants, stakeholders in the development of beef smallholder business models in Indonesia include government, academic and research organisations, business and the broader community (Figure 2). As mentioned above, three likely roles of government are as policy maker, research organisation, and extension service provider. The relevant policy agency is the Directorate General of Livestock and Animal Health Services (DGLAHS) within the Ministry of Agriculture. Its role is in formulating and implementing policies to increase production and productivity of livestock through input supply, animal health, marketing, processing, and capacity building. DGLAHS also works on various objectives for food self-sufficiency and operates programs such as the import of Australian breeding cattle and their distribution to various parts of the country in the Beef Self-Sufficiency Program (BSSP or in Indonesian PSDS).

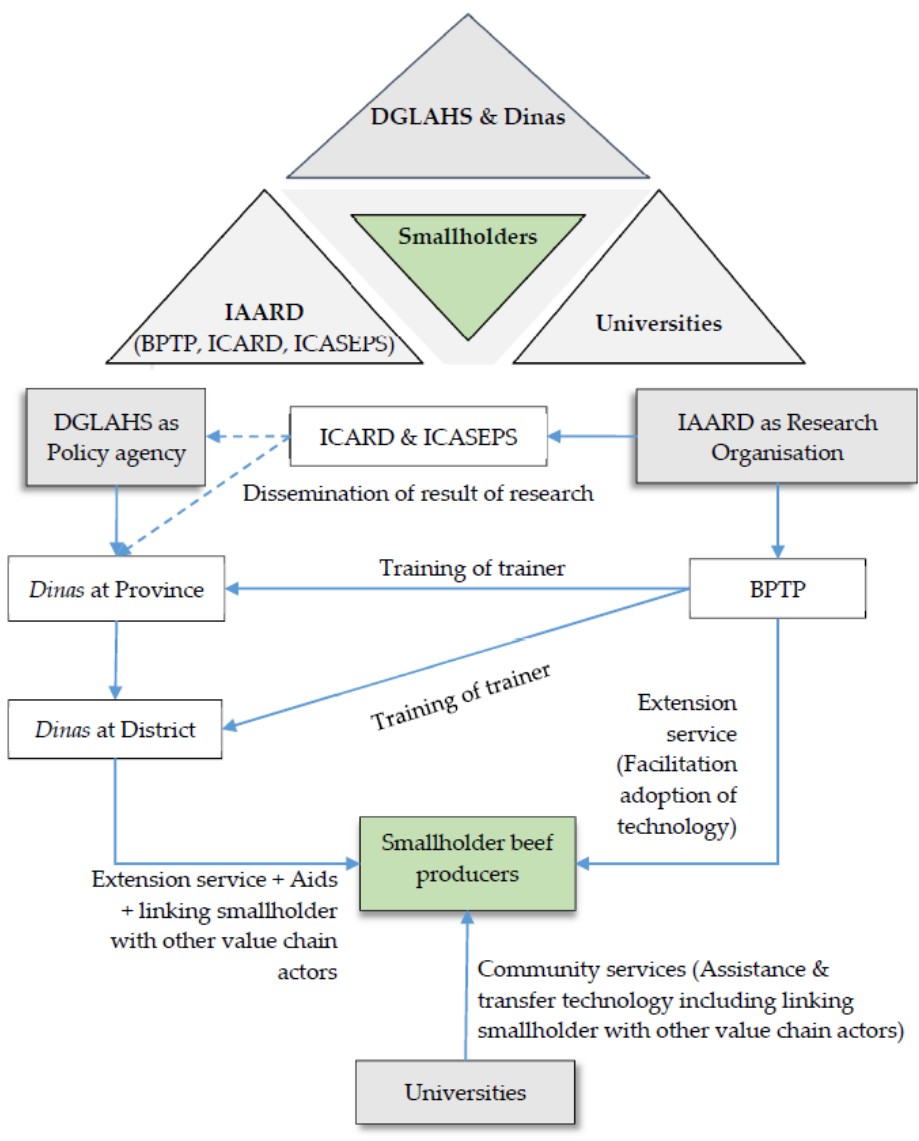

**Figure 2.** The role of external stakeholders.

The relevant research organisation is the Indonesian Agency for Agricultural Research and Development (IAARD), within the Ministry of Agriculture. It has two research centres related to the beef industry: The Indonesian Centre for Animal Research and Development (ICARD) and the Indonesian Centre for Agricultural Socio-Economic and Policy Studies (ICASEPS). They together formulate and help implement technical aspects of policies, including the dissemination of results of research.

Extension services are provided by local government (*Dinas* and the Assessment Institute for Agricultural Technology (AIAT/BPTP)). *Dinas* provides local agricultural extension services, while BPTP represents IAARD at the provincial level. BPTP facilitates adoption of appropriate technology. Additionally, BPTP has a role in training of trainers for *Dinas*. These actors promote awareness and knowledge of innovations among potential adopters. They can identify and support 'early adopters' to facilitate innovation uptake. A further expectation of extension services is that they link smallholders with other value chain actors, targeting inclusive business models.

Universities' roles are threefold: teaching, research, and community services. They provide education and training, assistance and support for field studies, and creation of new knowledge. Universities' role in community services would include to encourage smallholders' market participation and link with other value chain actors.

## 7. Conclusions and Research Directions

The Indonesian smallholder cattle system faces constraints in responding to booming domestic demand for beef. Current financial incentives undermine the viability of cattle breeding, despite the availability of new technologies and practices and the presence of facilitating agencies. This paper proposes a novel approach to the recognition of business models based on innovation behaviour in the Indonesian smallholder beef production and marketing system. It presents three components of business models: their value proposition, value architecture, and financing mechanism. It then maps them to innovation in pursuit of profitability. Business models that represent high levels of innovation adoption, market orientation, and crop-livestock integration are expected to show high profitability.

To occasion identification and analysis of such business models, the paper maps data and participation needs to business model development. Further, it identifies facilitation needs and the roles of external stakeholders in developing the business models and facilitating their uptake. Finally, the paper highlights the different typologies and elements of business models, as characterised by farming systems, economic, social and farmer characteristics. These findings are useful for identifying appropriate intervention strategies to accelerate the uptake of innovations in order to improve the productivity and sustainability of smallholder integrated crop-livestock farming systems in Indonesia.

**Author Contributions:** Conceptualization, Z.A. and D.B.; methodology, Z.A., D.B. and R.V.; literature review Z.A. and D.B.; formal analysis, Z.A., D.B. and R.V. and A.D.; investigation, Z.A. and A.D.; writing—original draft preparation, Z.A.; writing—review and editing, Z.A., D.B., R.V. and A.D.; project administration, Z.A., D.B., R.V. and A.D.; funding acquisition, Z.A. and R.V. All authors have read and agreed to the published version of the manuscript.

**Funding:** This research paper is completed while the lead author is a PhD.I candidate at the University of New England (UNE), Australia. The scholarship grant from this University and the financial support through the Australian Centre for International Agricultural Research (ACIAR) - UNE IndoBeef Project in conducting the field survey are hereby acknowledged.

**Conflicts of Interest:** The authors declare no conflict of interest.

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
