# Peer review of "Business Models and Innovation in the Indonesian Smallholder Beef Value Chain"

_sustainability, doi:10.3390/su12177020_

Round 1
Reviewer 1 Report
First: a few typing errors: line 75 and 146
Second: The paper is clearly written and describes the theoretical aspects of the appropriate business models. However, I expect the application of these models in the Indonesian smallholder cattle situation. Much more is known about the different stakeholders and their current attitudes in this beef production chain. See e.g.: Widi et al., 2020. https://journals.sagepub.com/doi/10.1177/0030727020915206
Third: In lines 48 - 53 the problem areas are identified and it is stated that constraints in these areas should be overcome by innovations. I expect that the business models indicate which innovations in these areas overcome the problems in the Indonesian smallholder situation.
Fourth: It would help all stakeholders by a few possible outlooks on the future beef production systems in Indonesia.
Conclusion: The application of the business models (see title of the paper) in the Indonesian smallholder cattle population is lacking.
Author Response
Dear Editorial Office, Sustainability
Thank you for the review and requests for revisions.
Please find attached our revised paper sustainability-888351. We also attach a table detailing our response to reviewers’ comments. The changes made in the revised paper are identified in yellow.
Yours sincerely
Zenal Asikin, on behalf of all authors.

Reviewer 2 Report
I commend the authors for a thorough review and discussion regarding application of business models for innovation in the Indonesian beef value chain. There are a few minor wording changes needed, but it is obvious the authors spent much time in preparation of concepts, review of literature, and organisation/presentation of the manuscript.
At some point, toward the end of the manuscript I wish the authors could have provided or speculated on the potential range in current currency values for cattle and/or beef carcasses as the industry exists now, versus what might be attainable through future applications of the business model components. This would have provided a more complete discussion.
What that being said, I have no problem with acceptance and publishing of the current manuscript.
Specific comments:
many places "per cent" is used vs. "%" symbol in others; recommend use of symbol everywhere consistently.
Line 78: maybe state livestock "serve as" a savings function, instead of "perform." Many times the term animal "performance" is used to represent the production trait level (i.e. if animals gained weight, they performed well, etc.)
Line 109: use authors' names to start sentence, not "[15]"
Line 120: some "sell" not "sells"
Line 308-309: include or specify "in Indonesia" for organization
Author Response

(The authors gave the same response as above.)

Reviewer 3 Report
- The authors should consider revising the title of the paper. It is too long and somewhat awkward; for instance, the element in the title reads: "cattle and beef value chain" means that there's difference between cattle value chain vs beef value chain? If the final use of the cattle is for beef, then it should be just be either "beef cattle value chain," "beef value chain,' or "cattle value chain."
- The feed cost for cattle has been estimated to be 60%, however, the author also mentioned that the smallholder cattle system uses on-farm resources rather than acquire the feed from outside. Would it be an overestimate of the cost of producing beef in the smallholder system? Which explains why it remains an attractive practice even though research typically suggest that the productively of such system is low.
- Line 36: "beef productivity" should be "beef production"; as productivity refer to per unit of input use while the sentence refer to the fact the production level could not keep pace with the increase in quantity demanded.
- Small holders resources could alleviate the burden of one concentrated grow area. When the world has turning into the small batch work is a diversified rather than the concentrated system, so the issue is about integrating the productions into smaller batching and each can be accomplished by each smallholder farms rather than concentrated in the intensive production unit. The diverse production units allow for more dynamic arrangement and allow for use the local resources. The unwillingness of such engagement might have something to do use the traditional concept for raring the livestock so that it is a form of physical saving rather than monetary saving so that they can "sell the cattle only when the household is in need of cash." While joining the integrated productive systematic means that they might receive the money not in the time they need it (might mean the value difference). When people do not trust the financial institution for saving and the falling purchasing power due to unstable currency might be the reason why smallholder farmers are not willing to keep the assets in terms o f currency. Has this been a probable reason observed in Indonesia?
Author Response

(The authors gave the same response as above.)

Round 2
Reviewer 1 Report
The paper is substantially improved. Title, aim and conclusions better reflect the content of the paper.